# A Cost-Effectiveness Analysis of Newborn Screening for Severe Combined Immunodeficiency in the UK

**DOI:** 10.3390/ijns5030028

**Published:** 2019-08-30

**Authors:** Alice Bessey, James Chilcott, Joanna Leaviss, Carmen de la Cruz, Ruth Wong

**Affiliations:** School of Health and Related Research, the University of Sheffield, Sheffield S1 4DA, UK

**Keywords:** severe combined immunodeficiency, SCID, cost-effectiveness, economic, newborn screening, neonatal screening

## Abstract

Severe combined immunodeficiency (SCID) can be detected through newborn bloodspot screening. In the UK, the National Screening Committee (NSC) requires screening programmes to be cost-effective at standard UK thresholds. To assess the cost-effectiveness of SCID screening for the NSC, a decision-tree model with lifetable estimates of outcomes was built. Model structure and parameterisation were informed by systematic review and expert clinical judgment. A public service perspective was used and lifetime costs and quality-adjusted life years (QALYs) were discounted at 3.5%. Probabilistic, one-way sensitivity analyses and an exploratory disbenefit analysis for the identification of non-SCID patients were conducted. Screening for SCID was estimated to result in an incremental cost-effectiveness ratio (ICER) of £18,222 with a reduction in SCID mortality from 8.1 (5–12) to 1.7 (0.6–4.0) cases per year of screening. Results were sensitive to a number of parameters, including the cost of the screening test, the incidence of SCID and the disbenefit to the healthy at birth and false-positive cases. Screening for SCID is likely to be cost-effective at £20,000 per QALY, key uncertainties relate to the impact on false positives and the impact on the identification of children with non-SCID T Cell lymphopenia.

## 1. Introduction

Screening newborns for severe combined immunodeficiency (SCID) using T-cell receptor excision circles (TRECs) on the blood spot was added to the core US Recommended Uniform Screening Panel in 2010 [1]. Whilst SCID screening has now been implemented in all US states, take up has been slower internationally, with a number of European countries conducting pilots [2,3]. Within the UK, the National Screening Committee (NSC) advises government and the National Health Service (NHS) concerning what newborn screening tests should be offered. This advice is based on independent assessments of the evidence of the potential benefits, harms and cost-effectiveness of each screening test [4].

There is strong evidence that diagnosis of SCID prior to symptomatic presentation improves the survival chances of SCID patients [5,6,7,8,9]. There is also evidence that earlier diagnosis substantially reduces the cost of treating SCID through reduced infection rates and subsequent hospitalisations [10,11]. Five economic analyses of SCID screening also suggest that screening may be cost-effective in the US, New Zealand and the Netherlands [12,13,14,15,16]. As there are differences between countries in terms of treatment costs and benefits and how health interventions are assessed, a UK-specific cost-effectiveness model was developed. 

The cost-effectiveness study reported here was undertaken on behalf of the UK NSC, to inform its consideration of screening for SCID within the NHS Newborn Blood Spot Screening Programme [4]. The study incorporates UK data on incidence and current practice for SCID treatment in the UK together with quality of life estimates. The study uses economic threshold analyses to examine the robustness of the results to the potential disbenefit of the high false-positive rate of the TREC test and the incidental identification of non-SCID T Cell lymphopenia (TCL) conditions.

## 2. Materials and Methods

A decision-tree cost-effectiveness model was developed to estimate the cost-effectiveness of including the additional TREC screening test for SCID in the NHS Newborn Blood Spot Screening Programme. A cost-effectiveness model provides the framework to enable evidence from a number of sources to be used together to estimate the total costs and health benefits for both the existing SCID pathway and for the pathway with the inclusion of newborn screening. The results of the model can be used to estimate the value of a new intervention, such as screening, using the incremental cost-effectiveness ratio (ICER).

The decision-tree model used is shown in Figure 1. Following a positive TREC screening test, patients undergo flow cytometry testing in order to confirm a diagnosis of SCID. Other clinical factors and genetic testing are used to differentiate the non-SCID TCL cases also identified by screening. Due to limitations in the evidence, and the focus on SCID patients, only the incremental costs associated with being identified via screening rather than symptomatically were included for the non-SCID TCL group. There are, therefore, no costs associated with non-SCID TCL patients in the no-screen arm. Life tables were used to estimate lifetime costs and health outcomes measured in terms of quality-adjusted life years (QALYs). The model takes an NHS and Personal Social Services perspective; costs were estimated at 2014/15 rates and costs and benefits were discounted at 3.5%. The annual number of births for the UK were estimated from the average of 10 years, 2005–2015 [17,18,19].

Patients with SCID identified through screening or those diagnosed early due to a family history in the no-screen arm are assumed to undergo early haematopoietic stem cell transplantation (HSCT), or in the case of Adenosine deaminase SCID (ADA-SCID), gene therapy, and have improved outcomes compared to those diagnosed and treated later due to symptomatic presentation [5,6,7,8,9].

Based upon the 82 cases referred to the two specialist SCID treatment centres in the UK between 2008 and 2012, a UK incidence of 1 in 43,600 (95% Confidence Interval (CI) 1:74,000, 1:29,000) was estimated. This includes a subjective estimate that around 1.5 patients with SCID may die annually before they are diagnosed. Based on the UK experience, 30% of patients were assumed to be diagnosed early due to a family history and 17% are assumed to have ADA-type SCID [20]. A full list of model parameters is given in Table 1.

A systematic search [31] identified five studies reporting on the comparative effectiveness of early vs late transplant [6,7,8,9,32]. The Brown et al. study [6] was selected as it is a UK-based study and was the only study to provide both pre- and post-HSCT mortality estimates. The model assumes that all deaths occur in the transplantation year and that surviving patients have a normal life expectancy. Patients with ADA-SCID have additional treatment options including enzyme replacement therapy prior to transplant and gene therapy for those with a non-matched donor [33]. ADA-SCID-specific parameters are given in Appendix A.

The baseline analysis assumed a cut-off of 20 copies/μL for the TREC screening test, with a presumptive positive rate of 0.04%, resulting in an estimated 312 referrals to flow cytometry per year [22]. Data from the Californian screening programme was used to estimate the incidence of non-SCID TCL conditions identified at screening due to the relatively low cut-off values used and the large number of births compared to other states [21,34]. Given that the estimated UK presumptive positive rate is higher than in California, it was assumed that the UK would have a higher proportion of cases being classed as false-positive rather than a higher incidence of other non-SCID T-cell lymphopenia [21,22]. 

While studies have demonstrated the TREC test to have perfect sensitivity at the expected TREC cut off [23], a sensitivity of 0.99 was used in the model to take into account the potential for false negatives in a population-based screening programme. The positive predictive value of the TREC test for SCID was estimated at 5.3%. 

The cost of the screening test kit is estimated at £3.50. The additional labour for the TREC test was estimated based on expert opinion to be 0.5 full-time equivalent of NHS grade-5 members of staff per each of the 13 laboratories, to give an additional cost of £166,000 per year, and equipping the laboratories was estimated at £35,000 excluding installation costs [35].

The patient pathway following a positive TREC test is shown in Figure 2. It is assumed that the flow cytometry test is able to differentiate healthy false-positive patients, SCID patients and some of the different TCL conditions. Markers such as physical features and the presence of other severe health problems can further differentiate non-SCID TCL patients [36]. Genetic tests differ depending on suspected conditions [27]. Only the incremental costs associated with being detected via screening, such as immunology appointments, are applied to the non-SCID TCL group. Longer follow-up costs are included where screening enables an earlier diagnosis, such as in the syndromes group, or where patients may not have presented symptomatically, such as the idiopathic TCL group. As some of the non-SCID patients are likely to be diagnosed at birth, these assumptions may overestimate the incremental costs.

Data on length of stay for patients diagnosed early and late was based upon current practice in one of the two UK SCID centres [24]. As shown in Table 1, the number of inpatients days were substantially reduced for those with a SCID diagnosis at birth compared to those with a later diagnosis. Total inpatient cost of HSCT was calculated from NHS reference costs [26] for elective and non-elective HSCT adjusted for the SCID specific length of stay. The Great Ormond Street Hospital (GOSH) data include readmittances and is assumed to include complications following transplants, such as additional transplants and graft versus host disease (GvHD), but does not include the length of stay at other hospitals before the patients were referred to GOSH. Full costing details, including the costs of longer term treatments such as immunoglobulin, are given in the Appendix A [28].

No studies were identified that report health-related quality of life utilities in SCID patients. To estimate quality of life, a clinician mapped information from a database of 78 UK SCID patients, transplanted between 1979–2015, onto the EuroQol 5-dimensions 3-level (EQ-5D-3L) health state descriptions [37,38]. The EQ-5D-3L time trade-off valuations were applied to the health state profiles to derive health utility estimates. The average health state utility for the 27 patients diagnosed at birth was 0.96 (sd 0.09) compared to 0.82 (sd 0.25) for the 51 later diagnosed patients. Figure 3 shows the cumulative distribution of the EQ-5D scores and the QALY dimension are shown in the Appendix A. Treatment for SCID has improved over the time period included, in order to explore this affect, only EQ-5D-3L estimates for patients between 2000–2015 were used in a sensitivity analysis. 

The systematic search [31] identified six studies that reported long-term clinical outcomes for SCID patients undergoing HSCT [7,39,40,41,42,43]. While it was not possible to synthesise the data; certain outcomes, such as the proportion of patients classified as healthy, low-weight, and developmental delay, were consistently different in the early vs late transplanted cohorts [7,42,43]. It is assumed that ADA-SCID patients undergoing either HSCT or gene therapy have the same long-term outcomes as general SCID patients. Full details of the long-term outcomes including costs are provided in Appendix A. 

Parametric distributions were used to characterise parameters in the model, see Appendix A, and 10,000 replicates were used to estimate uncertainty around the results. A number of one-way sensitivity analyses were also conducted. These included halving and doubling the incidence rate. Increasing the TREC cut-off rate to 30 copies/μL, with the increased presumptive positive cases assumed to be additional false-positive cases in the first instance and a proportional increase in the non-SCID TCL cases in the second. Increasing the cut-off rate does not change the sensitivity of the TREC screening test as in the Adam et al. [22] study no SCID cases had a level of about 15 copies/μL. However, a higher cut-off may be considered, at least initially, to comply with manufacturer’s recommendations or to ensure the sensitivity found in smaller studies is maintained in a nationwide screening programme [3]. The proportion of SCID patients identified due to a family history in the no-screen arm was varied, as was the cost of the screening test. To explore the uncertainty around the mortality estimates, two analyses were run that varied the odds ratio of the pre- and post-transplant mortality rates. The discount rate for public health intervention of 1.5% for both costs and benefits was used [44] and a discount rate of 3.5% for costs and 1.5% for benefits based on the NICE guidance for a sensitivity analysis were also conducted [45]. 

An expected value of perfect information (EVPI) analysis was undertaken using the Sheffield Accelerated Value of Information (SAVI) programme [46]. EVPI calculates the value of eliminating all uncertainty in the model and parameter EVPI identifies the value of eliminating uncertainty in each individual parameter. 

In order to explore the potential economic impact on those with a false-positive TREC test result and the impact of diagnosing otherwise healthy infants with non-SCID TCL, a threshold analysis was conducted. This analysis calculates the minimum per person health-related utility decrement that would cause the overall cost effectiveness of screening to increase above the thresholds of £20,000 and £30,000 per QALY. Based on the Californian screening programme, it was assumed that 33% (7 out of 21) of the non-SCID TCL would be healthy at birth and, therefore, included in the analysis [34]. 

## 3. Results

The base case results are shown in Table 2. The cost per QALY gained is estimated at £18,222 (£12,013, £27,763). The probability that screening is considered cost-effective is 65% and 99% at the £20,0000 and £30,000 per QALY thresholds, respectively. Overall uncertainty is presented in the cost-effectiveness plane shown in Figure 4. Cost associated with the immediate treatment and HSCT are lower in the screened arm; however, the long-term follow-up and management costs are higher than in the no-screen arm. While the per patient costs are lower, and the rates of complications are either the same or lower in the screened arm, the increase in survival increases the number of patients experiencing both complications and the general follow-up costs leading to overall higher costs.

Table 3 presents the sensitivity analysis results. The results indicate that the cost-effectiveness is substantially improved by a reduction in the TREC test cost, an increase in incidence, and the use of other discount rates. The results are also sensitive to the TREC cut-off and the proportion of SCID that is currently detected by a family history. Increasing the TREC cut-off reduces the probability that screening is cost-effective, from 65% to 58%, at £20,000 per QALY but only marginally, from 99% to 98%, at £30,000 per QALY. However, the impact on the substantial increase in false-positive cases, up from 266 in the base case to 880, and any arising disbenefit would need to be considered carefully. In terms of the mortality rates for the ICER to go above £20,000 or £30,000 per QALY, the pre-transplant mortality rate in the early diagnosed cohort would need to increase to 10% or 28%, respectively. For the transplant mortality rate, the mortality rate would need to increase to 17% and 36%, respectively, in the early diagnosed cohort.

The overall EVPI per person affected by the decision is estimated at £0.15 per person. Assuming an annual number of births affected by the decision of 780,835 and a decision horizon of five years, the overall expected value of removing decision uncertainty for the UK is estimated at £595,900. Research or data collection exercises costing more than this amount would not be considered a cost-effective use of resources. The single-parameter EVPI (EVPPI) demonstrates that the key uncertainties in the model relate to the underlying incidence of SCID in the population (EVPPI of around £80,000) and the cost of early transplantation, specifically the length of stay in non-critical care (around £65,000). The second largest group of parameters relate to the relative survival benefit from early versus late transplantation and the proportion detected due to a family history (all under £20,000).

The model estimates that there will be, on average, 266 (24,796) false positives and 31 (16, 50) babies identified each year with non-SCID TCL conditions, excluding pre-terms. It is estimated that eight of non-SCID TCL cases would present as healthy at birth and would not be diagnosed at this stage in the absence of screening [34]. This group is likely to include patients with ataxia telangiectasia and idiopathic TCL.

The results exploratory threshold analysis, shown in Table 4, calculates the minimum quality of life decrement that would need to be applied to each healthy-at-birth non-SCID TCL patient and false-positive case in order to increase the ICER over the cost-effectiveness threshold of £20,000 or £30,000 per QALY. As an example to aid interpretation, with a cost per TREC test of £3.50, if eight healthy-at-birth children had, on average, a disbenefit of one QALYs lost as a result of being identified with idiopathic TCL through screening, we would have to assume that each of the 260 false-positive infants also suffered a disbenefit of over six quality-adjusted days in order for the cost-effectiveness of screening to go over a £20,000 threshold with discounting at 3.5%. The results for the separate false-positive and non-SCID TCL analyses are shown in Appendix A. 

## 4. Discussion

This CEA of SCID screening is the first to include QALY estimates based on SCID patients, UK-specific data from the two UK SCID treatment centres, and an EVPI analysis and is the first to attempt to quantify the impact on non-SCID patients. The model estimated that screening for SCID in the UK would identify 17 (14, 22) newborns with SCID annually, prevent 6.3 (4.0, 9.7) SCID-related deaths with a total gain of 184 (118, 274) discounted QALYs. Screening will increase total discounted lifetime costs by an estimated £3.3 (£2.4, £4.5) million per year. The ICER for SCID screening is £18,222 per QALY gained, with 65% and 99% probability of this being considered cost-effectiveness at a threshold of £20,000 and £30,000 respectively. The results are in line with previously published cost-effectiveness studies that used QALYs [14,16].

The EVPI analysis estimates the value of future research and key uncertainties. Further data collection could be used to improve the data on incidence, mortality rates, length of hospital stay and proportion detected by a family history but given the small number of SCID patients diagnosed in the UK each year, there is still likely to be uncertainty in these estimates. One-way sensitivity analyses were conducted on these parameters to explore how robust the results are to changes in the parameter values. A change in the mortality rates does affect the probability that screening is cost-effective. However, assuming only a very small survival benefit at pre- or post-transplant (5.5% and 2.2%) for those early diagnosed, there is a still a 68% and 43% probability that screening is cost-effective at a £30,000 threshold. Parameters were also compared to data from other countries, for example, the GOSH length of stay data used is comparable to US and French studies [10,11].

The mortality estimates were taken from retrospective studies which may not reflect current outcomes [6]. Survival rates, especially for those diagnosed late, are likely to have improved over time due to a raised awareness of SCID and better supportive care [47]. They also may not reflect wider current practice, for example, in the UK rotavirus, a live vaccination, has been given at 8 weeks of age since 2013 and causes severe problems in SCID patients [48]. These impacts will not have been fully captured in the retrospective survival and outcomes estimates used. 

There are other uncertainties and factors that may not have been fully captured in the EVPI but may still affect the decision. These include the impact on false-positive and non-SCID TCL cases, the quality-of-life estimates, and choices on the cost of the test and the discount rate used. Sensitivity analyses showed that the cost of the test and the choice of discount rate had a sizeable impact on the ICER, with the probability that screening is cost-effective at £20,000 increasing from 65% to 95% when the cost of the TREC test is reduced from £3.50 to £2.50.

There is a lack of quality-of-life utility values that could be used for those with a false-positive or non-SCID TCL result. Whilst screening for untreatable conditions is generally not recommended, the majority of the non-SCID TCL cases identified are likely to present symptomatically without screening at or soon after birth [34]. It is assumed that for these patients, the marginal impact of the screening results is negligible. Earlier diagnosis in TCL-related conditions, for instance, Ataxia-Telangiectasia, has potential benefits and disbenefits for patients and families, with potentially ambiguous impacts on quality of life. Benefits may include a simplified diagnosis pathway, earlier provision of supportive treatments and allow families to prepare both practically and emotionally for the future. Conversely, in the case of conditions such as Ataxia-Telangiectaisa, earlier diagnosis may lead to premature medicalisation, with attendant stress and anxiety, together with the loss of a normal childhood [49,50,51,52]. 

The evidence from the Californian screening programme and the wider US shows some benefit from screening in non-SCID TCL patients. In California, of the 63 cases diagnosed with a syndrome, 16% received immune-directed treatments. Of the 47 cases of DiGeorge Syndrome, 9% received a thymus transplant while the other 90% had partial DiGeorge syndrome with improving T cell function over time [21]. In a separate study of patients diagnosed with 22q11.2 Deletion Syndrome, the most common cause of DiGeorge, 45% of patients were diagnosed prior to the screening result [53]. These results show that there may be some benefit to patients who would not have been diagnosed either symptomatically or through a family history prior to the TREC screening result. The inclusion of this benefit would improve the probability that SCID screening is cost-effective. However, more research, which includes a comparison with non-screen detected cases, is needed to fully quantify the potential impact on these patients. 

The greatest impact is likely to be on those defined as idiopathic TCL. The experience of this in California is that while some will resolve spontaneously, others will remain without a firm diagnosis [54]. The uncertainty in this diagnosis and potential for unnecessary medicalisation mean that there is likely to be an impact on the quality of life [52]. This study includes the first attempt to quantify the potential impact to those with a false-positive result and those who would be otherwise healthy-at-birth infants. The impact on these patients was raised as an issue in the Van der Ploeg study [16]. The threshold analysis calculates the number of quality-adjusted days that would need to be lost in order for the ICER to go above £20,000 or £30,000 per QALY. The disbenefit would have to be relatively large, in the order of one QALYs per healthy-at-birth non-SCID TCL and six quality-adjusted days per false-positive case. 

There is also an absence of quality-of-life utility evidence in the SCID population. This methodological weakness is not specific to SCID, but rather arises primarily from deeper foundational, if not philosophical, controversies associated with undertaking economic evaluation in child health [55]. Whilst there have been recent developments with the CHU9D and the paediatric EQ-5D-Y [56], there are currently no paediatric value sets available. In the absence of such measures, this study has used an adult EQ-5D valuation applied to health state descriptions generated with reference to medical records of affected children. This is the first attempt to measure the quality of life of SCID patients using a utility instrument without using proxy conditions. Whilst there are undoubted methodological weaknesses in this approach, there is no evidence to suggest a structural bias in the estimates obtained.

An additional factor that is not included in this analysis is the current timing of live vaccinations such as BCG and rotavirus. If live vaccinations are given before the screening results are known, then the benefits from screening may be reduced [57]. In the UK, it is recommended that the BCG vaccination should be given to high-risk infants or to all those born in areas with a TB incidence of over 40 in 100,000 on the neonatal ward [58]. The model assumes that no live vaccinations are given prior to the newborn screening results. However, the impact of changing the BCG vaccination recommendations has not been considered within this analysis. 

In 2016–2017, the UK NSC considered the establishment of SCID screening within the NHS Blood Spot Screening Programme. The economic study reported here comprised part of the evidence base considered by the Committee in forming its advice to ministers to evaluate SCID screening within a pilot study. The purpose of this evaluation being to address many of the uncertainties identified in this analysis including, the number of families with healthy babies who might be told they are ill when they are not (false positives), what care and treatment to offer babies with other causes of low numbers of white cells, how many babies are born have a prior family history and how the laboratory and treatment services will cope with the additional workload [59]. Following the completion of the pilot, the Committee will consider again whether SCID screening should be part of the wider Programme.

## 5. Conclusions

Screening for SCID is likely to be cost-effective at £20,000 per QALY, and is robust to changes to model parameters, key uncertainties relate to the impact of false positives and the identification of children with non-SCID TCL.

## Figures and Tables

**Figure 1 IJNS-05-00028-f001:**
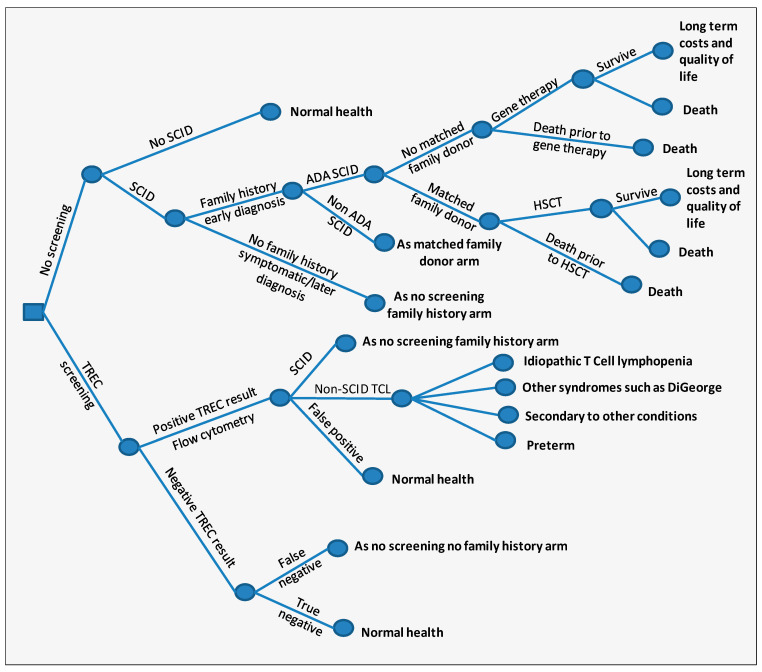
Model diagram. The figure represents the patient pathway in the screened and no-screened arm of the model. Some arms are clones of other arms; however, different parameter values are used in individual arms.

**Figure 2 IJNS-05-00028-f002:**
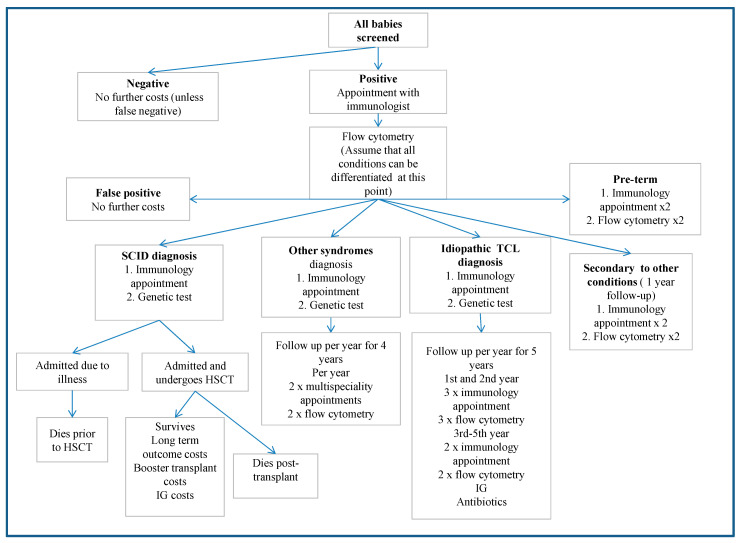
Screened patient pathway. The figure represents the pathway for all screened patients and includes the test and appointments for all non-SCID diagnosed patients.

**Figure 3 IJNS-05-00028-f003:**
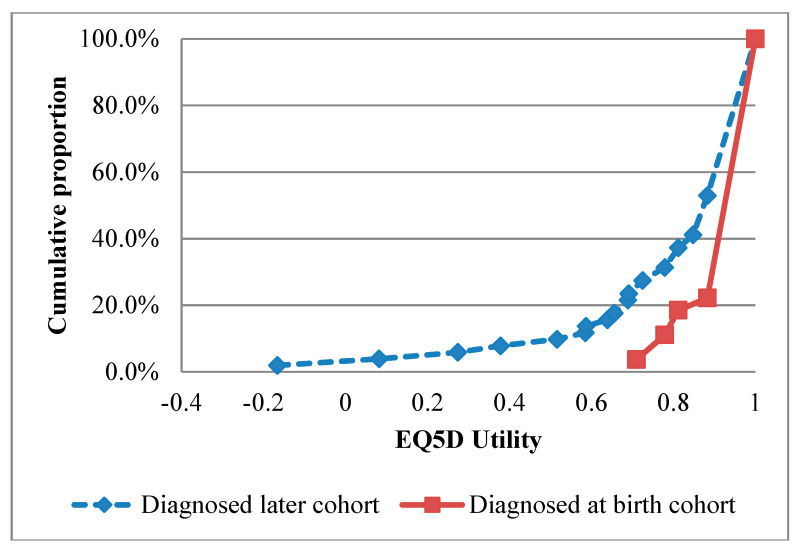
Cumulative distribution of EuroQol 5-dimensions (EQ-5D) utility scores for the diagnosed-at-birth cohort and the diagnosed later cohort. The red square line represents cumulative distribution of EQ-ED scores for the 27 patients in the diagnosed-at-birth cohort from Great Ormond Street Hospital. The blue diamond line represents the cumulative distribution of EQ-5D scores for the 51 patients in the diagnosed later than birth cohort.

**Figure 4 IJNS-05-00028-f004:**
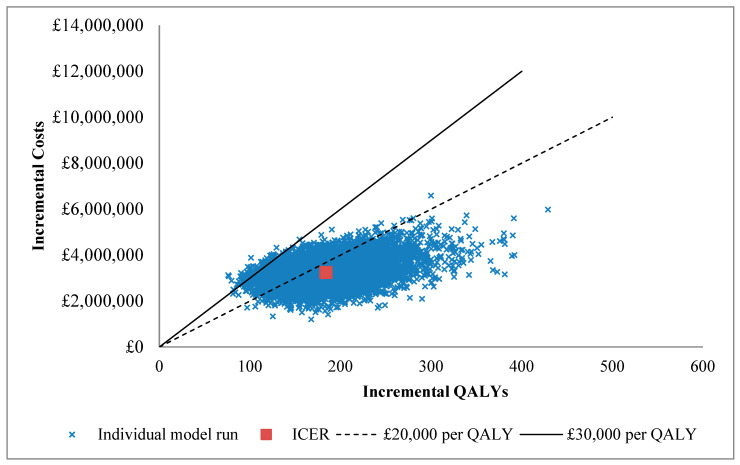
Cost-effectiveness plane. Each blue cross represents an individual model run. The red square represents the incremental cost-effectiveness ratio (ICER). The dotted line the £20,000 per quality adjusted life year (QALY) threshold hold and the solid line the £30,000 per QALY threshold.

**Table 1 IJNS-05-00028-t001:** Parameters table.

Parameter	Mean (95% Confidence Interval)	Reference
Number of births (UK)	780,835	[17,18,19]
Incidence of SCID	1:49,000 (1:39,857, 1:61,527)	[20]
Incidence of undiagnosed SCID	1:521,000 (1:167,052, 1:7,236,800)	[20]
Incidence of syndromes	1:45,000 (1:24,390, 1:110,606)	[21]
Incidence of secondary conditions	1:130,000 (1:50,686, 1:782,506)	[21]
Incidence of idiopathic TCL	1:99,000 (1:42,255, 1:432,482)	[21]
Incidence of positive TREC in pre-terms	1:99,000 (1:42,255, 1:432,482)	[21]
Presumptive positives (20 copies/µL)	0.041% (0.0035%, 0.1018%)	[22]
Sensitivity for SCID	0.99 (0.985, 0.998)	[23]
Proportion of SCID patients with a family history	0.30 (0.21, 0.41)	[20]
Proportion of SCID that is ADA-SCID	0.17 (0.1, 0.26)	[20]
Proportion of SCID patients with a matched family donor available	0.25 (0.07, 0.5)	[20]
Pre HSCT mortality (late diagnosed)	35.3% (22.8%, 49.3%)	[6]
Pre HSCT mortality (early diagnosed)	1.68% (0.11%, 7.63%)	[6]
HSCT mortality (late diagnosed)	38.7% (22.4% 56.3%)	[6]
HSCT mortality (early diagnosed)	8.48% (1.79%, 23.4%)	[6]
Number of days HSCT	54.0	[24]
Early diagnosis—Total days non-critical care	82.6 (50.3, 122.8)	[24]
Early diagnosis—Total days critical care	3.96 (0.15, 8.41)	[24]
Late diagnosis—Total days non-critical care	144 (108.6, 184.3)	[24]
Late diagnosis—Total days critical care	8.19 (3.72, 14.4)	[24]
QALYs—early diagnosis (1979–2015 cohort, base case values)	0.95	[25]
QALYs—late diagnosis (1979–2015 cohort, base case values)	0.82	[25]
QALYs—early diagnosis (2000–2015 cohort)	0.96	[25]
QALYs—late diagnosis (2000–2015 cohort)	0.87	[25]
Cost HSCT (early diagnosed)	£128,363	[24,26]
Cost HSCT (late diagnosed)	£231,186	[24,26]
Cost death prior to HSCT	£43,368	[24,26]
Presumptive positive cost	£276	[24,26]
Diagnosis SCID	£711	[26,27]
Diagnosis idiopathic SCID and syndromes	£1,551	[26,27]
Idiopathic SCID follow up (5 years discounted)	£20,142	[26,28,29,30]
Syndromes 4 year follow up	£4,872	[24,26]
Follow up preterm & secondary to other conditions	£533	[24,26]

UK—United Kingdom; SCID—severe combined immunodeficiency disorder; TCL—T Cell lymphopenia; TREC—T cell receptor excision circle; ADA—adenosine deaminase deficiency; HSCT—hematopoietic stem cell transplantation; QALYs—quality-adjusted life years.

**Table 2 IJNS-05-00028-t002:** Results.

		Screening	95% CI	No Screening	95% CI	Incremental	95% CI
Outcomes	SCID cases diagnosed symptomatically	0.2	(0.1, 0.3)	11.0	(8.4, 14.1)	−10.9	(−13.9, −8.2)
	SCID cases diagnosed via a family history	0	(0, 0)	4.9	(3.1, 7.0)	−4.9	(−3.1, −7.0)
	SCID cases not diagnosed	0	(0, 0)	1.5	(0.1, 4.8)	−1.5	(−0.1, −4.8)
	SCID cases screen detected	17.2	(13.5, 21.7)	0	(0, 0)	17.2	(13.5, 21.7)
	ADA SCID	2.9	(1.6, 4.7)	2.9	(1.6, 4.7)	0	(0.0, 0.0)
	SCID mortality	1.7	(0.6, 4.0)	8.1	(5.3, 12.0)	−6.3	(−9.7, −4)
Screening outcomes	Non SCID TCL	31.1	(16.3, 50.1)	0	(0, 0)	31	(16.3, 50.1)
	Pre-term	8.0	(1.9, 18.7)	0	(0, 0)	8	(1.9, 18.7)
	Total presumptive positives	322.1	(79.5, 852.9)	0	(0, 0)	0	(79.5, 852.9)
Costs	Direct screening costs	£3.04m	(£2.97m, £3.19m)	£0.00m	(£0.00m, £0.00m)	£3.04m	(£2.97m, £3.19m)
	Diagnosis and follow up pre-terms	£4,394	(£1,029.36, £10,311)	£0	(£0.00m, £0.00m)	£4,394	(£1,029.36, £10,311)
	Diagnosis and follow up non SCID TCL	£0.27m	(£0.12m, £0.51m)	£0.00m	(£0.00m, £0.00m)	£0.27m	(£0.12m, £0.51m)
	Diagnosis costs SCID	£14,235	(£11,112, £17,967)	£13,023	(£13,023, £15,954)	£1,212	(£61.7, £3,919)
	SCID treatment up and including HSCT	£3.35m	(£2.20m, £4.83m)	£3.63m	(£3.63m, £2.65m)	−£0.28m	(−£1.10m, £0.62m)
	SCID long-term costs	£2.03m	(£1.20m, £3.12m)	£1.30m	(£1.30m, £0.78m)	£0.72m	(£0.23m, £1.37m)
Totals	Total costs	£7.30m	(£5.96m, £9.06m)	£3.96m	(£3.96m, £2.92m)	£3.34m	(£2.36m, £4.47m)
	Total QALYs	410.1	(308.3, 527.3)	226.9	(226.9, 164.9)	183.17	(109.3, 267.2)
	ICER	£18,222	(£12,013, £27,763)				

CI—Confidence Interval; SCID—Severe Combined Immunodeficiency; ADA SCID—adenosine deaminase deficiency SCID; TCL—T-cell immunodeficiency; HSCT—hematopoietic stem cell transplant; QALYs—Quality-adjusted life years; ICER—Incremental cost effectiveness ratio.

**Table 3 IJNS-05-00028-t003:** Sensitivity analyses results.

Sensitivity Analysis		Screening		No Screening		Incremental			Probability Cost-Effective at Threshold	
		Costs	QALYs	Costs	QALYs	Costs	QALYs	ICER	£20k	£30k
Base case		£7.30m	410	£3.96m	227	£3.34m	183	£18,222	65%	99%
Incidence	Halved	£5.49m	223	£2.01m	113	£3.48m	110	£31,647	2%	35%
	Doubled	£10.97m	785	£7.89m	453	£3.07m	332	£9,266	100%	100%
TREC cut off	Additional false positives	£7.53m	412	£3.99m	227	£3.54m	185	£19,137	56%	98%
	Increase proportionately	£7.44m	409	£3.98m	227	£3.46m	182	£18,983	56%	98%
Family History	10% family history	£7.32m	411	£4.06m	183	£3.26m	229	£14,237	95%	100%
	50% family history	£7.30m	410	£3.89m	269	£3.42m	141	£24,208	14%	84%
Test cost	£1.50	£5.75m	410	£3.98m	227	£1.77m	183	£9,674	100%	100%
	£2.50	£6.55m	413	£3.99m	228	£2.56m	184	£13,876	95%	100%
	£4.50	£8.10m	410	£3.98m	227	£4.11m	183	£22,471	25%	92%
Discount rate	1.5% both costs and health benefits	£8.82m	702	£5.02m	391	£3.80m	311	£12,219	99%	100%
	1.5% health benefits & 3.5% costs	£7.32m	703	£4.00m	392	£3.32m	311	£10,680	100%	100%
QALYs	2000–2015 cohort QALYs used	£7.32m	415	£3.98m	235	£3.34m	180	£18,588	61%	98%
Pre-transplant mortality	Early diagnosed mortality (OR)									
	8.15%(0.23)	£7.12m	381	£3.67m	206	£3.45m	175	£19,691	51%	97%
	29.40%(0.83)	£6.44m	294	£3.48m	182	£2.95m	113	£26,237	12%	68%
Post-transplant mortality	Early diagnosed mortality (OR)									
	17.42%(0.45)	£7.16m	369	£3.68m	203	£3.48m	166	£20,915	39%	96%
	36.77%(0.95)	£6.79m	283	£3.58m	178	£3.21m	105	£30,746	3%	43%

QALYs—Quality-adjusted life years; 20k—£20,000; 30k—£30,000; m—million; TREC—T-cell receptor excision circles; OR—odds ratio.

**Table 4 IJNS-05-00028-t004:** Threshold results.

Healthy at Birth Disbenefit (QALY)	Cost per TREC Test £3.50	
	False-Positive Disbenefit Threshold (Quality-Adjusted Days)	
	Discounting 3.5%	Discounting 1.5%
	Mean (95% CI)	Mean (95% CI)
0	17 (0, 22)	171 (1,113, 91)
1	6 (0, 14)	159 (1,069,83)
2	0 (0, 6)	148 (1,025, 75)
3	0 (0, 0)	136 (980, 67)
4	0 (0, 0)	125 (936, 59)
10	0 (0, 0)	55 (672, 11)
20	0 (0, 0)	0 (230, 0)

TREC—T-cell receptor excision circles; QALYs—Quality-adjusted life years, CI—Confidence interval.

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
