# Peer review of "A Cost-Effectiveness Analysis of Newborn Screening for Severe Combined Immunodeficiency in the UK"

_2409-515X, 2019, doi:10.3390/ijns5030028_

Round 1

Reviewer 1 Report

Major issues

The overall process is not clear. I should read up to line 83 to understand that there is another test between TREC and HSCT (flow cytometry [testing], FCT) and lines 91-92 to understand that the whole screening process represented in Figure 1 is made of TREC and FCT.

Additionally, the TREC has a presumptive positive rate of 0.04% as in reference 22 – check table 1 which reports 0.0034% instead. Different cut-offs increase the presumptive rate up to 1% as I read in 22. It is not clear how the sensitivity is affected by the different cut-offs being already at 0.99 with the 20 copies/microliter cut-off as reported in table 1. 

Moreover, is the FCT done on TREC positive subjects a gold standard? in this case it should be clearly stated because, even if the results don’t change (false positive to TREC are identified with FTC), it has an influence on the discrimination power of the screening… in my opinion, in-fact, the screening is done with the TREC, the diagnosis with the FTC and, then, subjects positive to FTC  undergo HSCT (as reported, for example, in “Screening for severe combined immunodeficiency in neonates. Clin Epidemiol. 2013; 5: 363–369”). 

So, when you refer to the screening, you should intend the TREC alone. This test needs a confirmatory assessment (FCT) due to its very low PPV with respect to SCID (18.8 / 312 = 6% if I use your number with a 1:40,000 SCID incidence).

The discrimination power of the TREC is:

-      Sensitivity - 0.99

-      Specificity– 0.99 (as calculable from the data you provided and, of course, strongly pushed up by the very low SCID incidence).

Minor issues.

-      What happens in case of incidental identification of non-SCID TCL conditions? 

-      ADA-SCID management is not shown in the model. Please explain your choice and describe, at least in the text, what happens to ADA-SCID patients.

-      Abstract and line 46: Explode the TCL acronym

-      46: which are the consequences of “incidental identification of non-SCID TCL conditions?

-      49: for costs purposes, are you talking of an additional examination from the already collected newborn blood spots?

-      59: a description of the screening/diagnostic process (TREC + FCT) could be added before this line.

-      62-63: “Incremental costs associated with identifying patients with non-SCID TCLs are included in the model”. It is not clear which are these incremental costs and what happens to those that are not identified (no screening arm).

-      65: you report “UK [SCID] incidence of 1 in 40,000”, but in table 1 (line 3) you report “Incidence of SCID = 1:49,000”. Which is the correct value? I cannot move from 1:40,000 to 1:49,000 even if considering the 1,5 average annual SCID newborns that dies before diagnosis. 

-      Figure 1 –

o  If you accept the major issues you should change it because you have represented the whole screening process (TREC+FCT) – but this representation is also not clear. False positives are only a theoretical possibility as the specificity is 1. Nevertheless, if you report them, you should also clarify the difference with true negatives (currently you have reported normal health for both, which is true as health state but not in terms of resource consumption as they should undergo some kind of treatment). Please clarify. 

o  Why Non-SCID TCL is present only the “screen” arm? If you can diagnose another disease, this represents a benefit and a cost in the arm of the screening but it is also a BIG COST and a BIG DISBENEFIT in the arm of no-screening due to the rising costs and health problems linked to its late detection. 

o   I assume that “As no screening family history” and “As no screening no family history” are 2 clones. You should put clones after their master, so either invert the main arms (“no screening” first) or put the clones in the “no screening” arm and the masters in the screening arm.

o  In the “no screening” arm there are the paths after “non family” and “family” that are identical (I assume with different value of variables). Why don’t you clone the latter?

-      Table 1

o  You should add cost parameters too (a summary of what reported in the supplemental material).

o  Please add a reference for the sensitivity or reason for this assumption.

o  Where are “idiopathic SCID and syndromes not diagnosed at birth” in Figure 1? 

o  QALYs: what is the difference of the 2000-2015 cohort? which QALY lines are you using in the baseline model?

o  As said, the presumptive positive rate is 0.0034% in the Table 1 but 0.04% in line 83.

-      257-259: the sentence is not clear. What is “Respectively” referred to?

-      273: you should add “instead of £3,50”

-      305-306: remove the third “are”

-      318-319: I don’t understand what the “which” refers to

-      Table S3, the reference for the presumptive positive is different from the one in Table 1. I presume it should be 21 and not 20. The same “1 number error” happens for the “Number of days HSCT” (it should be 4 and not 3 in the supplemental material). Please check all the references in the supplemental material as it should be a systematic error.

Author Response

We would like to thank the reviewer for their detailed comments. We have made a number of changes to the manuscript based on them. Please see the attachment for the full response to the comments.

Reviewer 2 Report

Review of “The Economics of newborn screening for Severe Combined Immunodeficiency disorder” submitted to The International Journal of Newborn Screening.

Overall:  This is a significant study prepared as a prospective evaluation of SCID newborn screening in UK.  Previous studies from US, New Zealand, and the Netherlands suggest that SCID screening may be cost-effective.  This study uses incidence, standard practice, and cost data from UK to derive specific estimates of cost-effectiveness measures as the incremental cost effectiveness ratio.

While I agree with the overall conclusion, I find that some of the underlying assumptions and estimates tend to underestimate the effectiveness of SCID screening.  Some of the sources of my concern are the large number of UK parameters that are estimated from personal communication.  One significant parameter in this group is the incidence of undiagnosed SCID in the absence of newborn screening.  In addition, some of the estimates of the relative costs and consequences of early vs. late HSCT come from older studies, which may not reflect current practice.

The sensitivity analysis focuses in the direction of making the screening less cost-effective, so as to estimate the threshold for screening implementation.  This is important, but needs to be underlined so that readers do not take away the message the SCID screening may not be effective in other contexts.

Further, the study disregards any benefit from the identification of T-cell lymphopenia in syndromes and secondary conditions.  The choice of Ataxia-Telangiectasia as an epitome of this category is perhaps the worst-case example.  Perhaps 22q11.2 deletion syndrome could supply a more balanced discussion of this category.  

Disregarding the benefit of avoiding live vaccine in the entire category of TCL is a potential concern.

Finally, since early death in unscreened SCID provides significant cost savings, changes in the mortality in this group may be important.

Some specific comments:

Title:  I would prefer the title to include some indication that this evaluation is prospective and that it is tailored to UK.

Table 1:  Is the point estimate for the presumptive positive rate wrong?

Table 3:  There is a mix of one-way and two-way sensitivity assumptions.  Perhaps some of the more extreme examples (for example the second pre- and post-HSCT mortality rates in the early diagnosed cohort) and replace them with two-way scenarios (for example the TREC cutoff).

Table S1: The multispecialty follow up appointments for Syndromes should be included in the No Screening costs

Table S2:  The lack of difference between the early diagnosed and late diagnosed group for IG is based on what may be out-of-date data.

Table S3:  Are the beta distributions for the various HSCT mortality correct:  For example, for HSCT mortality in the late diagnosed group, isn’t the mean of beta(19:12) equal to 0.61?  Some notation of how these are used should be included.

Author Response

We would like to thank the reviewer for their comments. We have made a number of changes to the manuscript based on them. Please see the attachment for a full response to the comments.

Round 2

Reviewer 1 Report

My concerns were adequately addressed with revisions. 

Reviewer 2 Report

The revisions have significantly improved the paper.